# LeYOLO: Lightweight, Scalable and Efficient CNN Architecture for Object Detection

## Abstract

Computational efficiency in deep neural networks is critical for object detection, especially as newer models prioritize speed over efficient computation (Parameters and FLOP). This trend is evident in the latest YOLO architectures, which focus more on speed at the expense of lightweight design. This evolution has somewhat left lightweight architecture design behind for object detection applications. Unlike speed-oriented object detectors in the literature, SSDLite and low-parameters/FLOP-oriented classifier combinations are the only proposed solutions, leaving a gap between YOLO-like architectures and lightweight object detectors. In this paper, we pose the question: *Can an architecture optimized for parameters and FLOPs achieve precision comparable to mainstream YOLO models?* To explore this, we introduce LeYOLO, an efficient object detection model, and propose several optimizations to enhance the computational efficiency of YOLO-based models. This approach bridges the gap between SSDLite-based object detectors and YOLO models, achieving high precision in a model as lightweight as MobileNets. Our novel model family achieves a FLOP-to-accuracy ratio previously unattained, offering scalability that spans from ultra-low neural network configurations ( < 1 GFLOP) to efficient yet demanding object detection setups ( > 4 GFLOPs) **with 25.2, 31.3, 35.2, 38.2, 39.3 and 41 mAP for 0.66, 1.47, 2.53, 4.51, 5.8 and 8.4 FLOP(G)**.

## 1 Introduction

Initially introduced by Redmon et al. (2016), YOLO models are known for their object detection speed. These models have seen significant improvements in neural network architecture in recent years, taking advantage of modern computing power like GPUs. Essentially, YOLO models feature a backbone like that of classifiers, a neck that aggregates multiple levels of semantic information, and a head that refines detections across these levels. Detections are made on a grid, with spatial information greatly reduced, meaning detection boxes are aligned pixel by pixel.

Despite their inherent speed, there has been a noticeable shift in the development of YOLO models in recent years Jocher et al. (2022); Li et al. (2022); Wang et al. (2023); Jocher et al. (2023); Wang et al. (2024). With rapid advancements in GPU capabilities and new architectural innovations, the focus has shifted from lightweight models to those prioritizing speed. Consequently, YOLO models have become significantly faster despite increased parameters and FLOP [1] [2].

---

[1] We describe floating point operations as **FLOP**, defining all the number of arithmetical operations the neural network requires to perform inference.

[2] In our paper, 1 FLOP is roughly 2 MADD or 2 MACC. Thus, the variation in benchmarks such as MobileNet differs from their original paper.

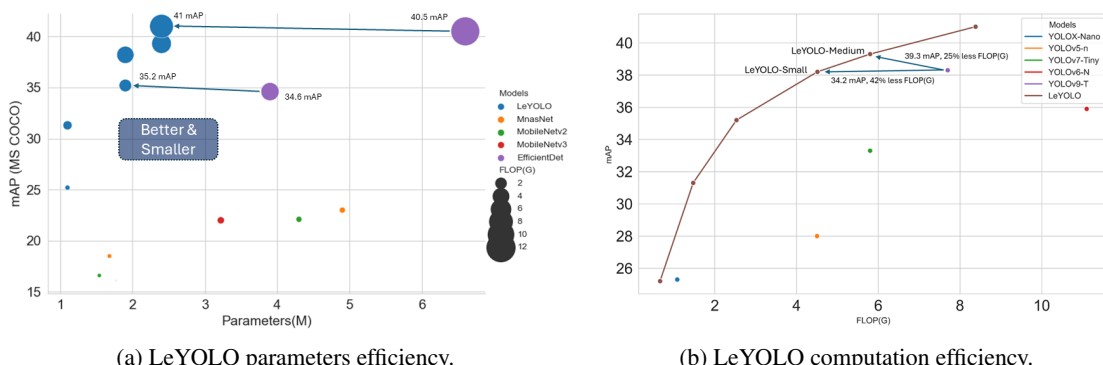

(a) LeYOLO parameters efficiency.

(b) LeYOLO computation efficiency.

Figure 1: LeYOLO compared to other sota object detection with low-parameters (a) and the sota mainline of low-cost YOLO (b)

On the other hand, research into optimizing parameter counts and computational costs has produced noteworthy models like MobileNets Howard et al. (2017); Sandler et al. (2018); Howard et al. (2019) and EfficientNets Tan & Le (2019; 2021). While these models are remarkable, they are primarily recognized for their exceptional classification abilities rather than object detection. Research has mainly focused on lightweight classifiers with optimized parameters in object detection, often paired with an object detection addon like SSDLite. Through our study of classification and object detection architectures, we have identified a research gap: *there is a lack of focus on optimizing architectures based on parameter counts and FLOP in the space between state-of-the-art fast object detectors and lightweight classifiers.* This gap leaves researchers with limited options, often leading them to rely on SSDLite Liu et al. (2016).

Fortunately, novel YOLO-based architectures embrace efficient computation, focusing on FLOP and parameters efficiency Moosmann et al. (2023); Fang et al. (2020); Ge et al. (2021); Yang et al. (2022); Hajizadeh et al. (2023); Wang et al. (2020).

Research in the object detection community is divided into three primary goals: inference speed, precision, and optimizing the precision (mAP) to computational cost (FLOP) ratio. From the previous statement, we raise several questions.
*Why don't speed and FLOP always correlate?* FLOP alone is insufficient, as it overlooks crucial inference factors like memory access cost, parallelism, and platform characteristics. The latest YOLO models Wang et al. (2023); Jocher et al. (2023); Wang et al. (2024) illustrate this by achieving faster speeds despite higher FLOP usage. Nevertheless, given the strong correlation with parameters, we still view FLOP as a valuable indicator of parameter efficiency in deep neural networks. However, we value FLOP for **parameter efficiency** and its valuable capacity to show a throughput correlation pattern with a decreasingly powerful embedded device (See Figure 2).
*Why focus on FLOP/parameter efficiency?* There are several reasons. Efficient models often lead to faster inference, which isn't always guaranteed due to inherent weaknesses in FLOP

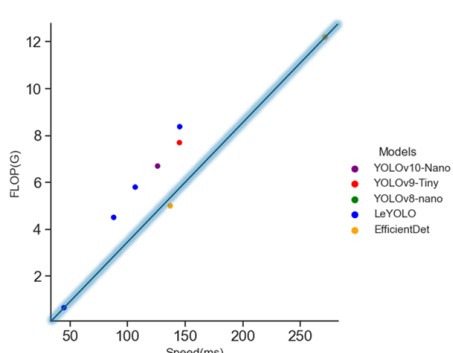

Figure 2: Flop-to-speed comparison on Jetson TX2.

measurements. Models with fewer parameters consume less power, making them ideal for battery-operated devices. Smaller models reduce the required bandwidth in scenarios where models are transmitted over a

network. Moreover, simpler models with fewer parameters are often more interpretable, helping us better understand neural network architectures. While deep neural networks are robust at generalizing functions and solving complex problems, crafting highly efficient architectures may offer insights into improving performance and understanding. We place significant importance on the number of parameters, even though recent studies have focused on maximizing execution speed, sometimes at the expense of parameter efficiency. Despite the reasons outlined previously that highlight other factors affecting execution speed, our FLOP-based study has enabled us to develop a faster object detection model that uses fewer parameters and achieves significantly higher accuracy than the state-of-the-art lightweight classifiers.

Furthermore, we demonstrate that our contribution competes with YOLO models at comparable scales. Our work proves that it is possible to optimize neural network architecture for object detection by proposing a new scaling approach that sits between lightweight classifiers and YOLO models, which are considered lightweight in terms of parameters and FLOP count. We present **LeYOLO**, a conceptually simple yet efficient architecture that embraces computationally efficient components for object detection, following the cores of EfficientNets Tan & Le (2019; 2021), MobileNets Sandler et al. (2018); Howard et al. (2017; 2019). Prioritizing efficient scaling, LeYOLO demonstrates superior performance across a broad scope of neural networks, surpassing ultra-low networks (less than 1 FLOP(G)), mid-range networks (between 1 and 4 FLOP(G)), and even models exceeding 4 FLOP(G) as Figures 1a and 1b shows.

In all its versions, LeYOLO achieves 25.2%, 29%, 31.3%, 35.2%, 36.4%, 38.2%, 39.3%, and 41% mAP on the MSCOCO validation dataset for 0.66, 1.126, 1.47, 2.53, 3.27, 4.51, 5.8 and 8.4 FLOP(G) and . In comparison, LeYOLO notably outperforms EfficientDet with 22% less inference time, 38% fewer parameters, and 13.6% increased performance. On mobile devices, LeYOLO outperforms the sota of object detectors from YOLO mainline and lightweight classifiers combined with SSDLite, as seen in Figure 2.

## 2 RELATED WORK

Our work focuses on finding an optimal architecture for object detection. We have combined two approaches: object detectors optimized for speed and low-cost classifiers that use well-established techniques to reduce the number of parameters.

**Lightweight classifiers.** LeYOLO draws inspiration from elements known for their high efficiency in optimizing the number of parameters. Notably, we leverage the performance of inverted bottlenecks, initially designed in MobileNetv2 Sandler et al. (2018) and later refined by the EfficientNet Tan & Le (2019; 2021) and GhostNet Han et al. (2020); Tang et al. (2022) families. Inverted bottlenecks, pointwise Lin et al. (2013), and depthwise convolutions are critical in architecture optimization. They play a crucial role in algorithmic contributions like MNASNet Tan et al. (2019), and many recent studies focused on hybridization, which aims to reduce the number of parameters in architectures utilizing self-attention, such as MobileViT Mehta & Rastegari (2022a) and others Wadekar & Chaurasia (2022); Mehta & Rastegari (2022b); Vasu et al. (2023). In our paper, we demonstrate that it is still possible to achieve a higher level of optimization, particularly in object detection, by further refining the use of inverted bottlenecks.

**Lightweight object detectors.** Originally designed to reduce the cost of object detectors by leveraging the widely respected VGG Simonyan & Zisserman (2014) feature extractor, SSD Liu et al. (2016) is an object detector closely related to the early YOLO models Redmon & Farhadi (2018). With the rise of low-cost classifiers, it became necessary to contribute to this growing field, leading to the creation of SSDLite, an optimized version of SSD primarily based on the principles of MobileNets with grouped convolutions. Since then, there hasn't been a method that surpasses this approach. However, SSDLiteX Kang (2023) has emerged as an attempt to improve SSDLite's performance on MobileNets. On the YOLO side, other researchers have also explored optimizing the mainline by incorporating the mentioned elements. Tinyssimo YOLO Moosmann et al. (2023) seeks to reduce overall costs by building on the earliest YOLO architectures Redmon et al. (2016). While the optimization is promising, it doesn't quite compete with even the lowest

scaling levels of YOLO or classifiers combined with SSDLite. Similarly, other studies Fang et al. (2020); Yang et al. (2022); Hajizadeh et al. (2023); Wang et al. (2020) have used lightweight classifier elements like depthwise convolutions and older techniques such as fire modules Iandola et al. (2016) to reduce the number of parameters. More recently and notably, YOLOX Ge et al. (2021) and YOLOv9 have offered a solid alternative to their base scaling by presenting a very lightweight model in terms of parameter count. YOLOX achieves this by using depthwise convolutions instead of standard convolutions, with a kernel size greater than three and a reduced image size. As for YOLOv9, we are exceptionally inspired by the authors' significant contribution to the parameters and information optimization field. YOLOv9 keeps an optimization to a mainline YOLO scaling level, discarding mobile-oriented and low-crafted-parameters-oriented neural networks. There is more discussion on the state-of-the-art deep neural network in Appendix A.3.1.

**Discussion on contribution.** We introduce LeYOLO, a model inspired by architectures renowned for parameter optimization. LeYOLO proves that optimizing low-cost object detectors is possible, offering a strong alternative to the lowest scalings of YOLO models. It also provides a significant performance boost compared to the state-of-the-art object detectors in downstream tasks, largely reliant on the well-known SSDLite. Our study offers a highly modular approach compared to existing YOLO solutions and state-of-the-art lightweight classifiers, with YOLOv9 as an example of a comparable study at a larger scale. Our focus is on "mobile" or lightweight neural networks, so our proposal is less optimized for block acceleration and parallelization techniques, as seen in YOLOv7. Also, the extensive use of depthwise convolution, as presented in Chapter 3.1.1, might reduce the throughput of our solution, as first introduced in the ShuffleNet Ma et al. (2018) study.

LeYOLO brings several optimization and contributions: **(i) Better classifier downstream performance.** For a given parameter scaling budget, LeYOLO outperforms state-of-the-art low-cost classifiers combined with SSDLite by reducing the number of parameters and increasing the precision of MSCOCO. Adding LeYOLO neck and head to sota low-parameters and cost-oriented backbone enables better throughput, accuracy, and much lower parameters for several scaling. **(ii) Tiny scaled YOLO alternative.** LeYOLO, with its optimized backbone, neck, and detection head, surpasses nano and tiny-scaled mainline YOLO neural networks on object detection. The architectural choice of the LeYOLO backbone proves its superiority compared to other low-cost backbones combined with the LeYOLO neck and head, being better at scaling and accuracy-to-parameters and FLOP ratio. **(iii) Improved throughput.** LeYOLO gets better throughput than sota object detector with low-oriented parameters thanks to its optimized architecture. Both complete LeYOLO and downstream tasks on low-cost classifiers get better throughput. However, LeYOLO reaches its peak strength on mobile, embedded, or low-powerful devices, getting closer to what we seek in terms of parameter efficiency: enabling the power of reliable object detectors directly to inference on small devices, having few parameters to share within clusters of peripheral devices, bringing YOLO closer to edge AI step by step.

## 3 LeYOLO: A low-parameters oriented object detector

### 3.1 LeYOLO Architecture

#### 3.1.1 LeYOLO Block.

Regarding parameter and mAP efficiency, surpassing architectures based on depthwise and pointwise convolutions is challenging. However, our experiments with the inverted bottleneck block found that optimizing the number of channels can significantly reduce computational demands, particularly at large spatial feature map sizes. With careful optimization, the initial pointwise convolutions may even be optional, leading to a drastic reduction in the number of parameters and FLOPs in the early layers of the neural network, with minimal impact on accuracy - leading to lower cost scaling, especially at high spatial size. The core block of LeYOLO is primarily designed to reduce the number of parameters based on an input $x \in \mathbb{R}^{B,C,H,W}$. Our

block applies a $1x1$ convolution followed by an $n \times n$ convolution. Finally, a third $1x1$ convolution is used to project or restore the feature map to its original number of channels. The pointwise convolutions mainly project the feature maps at different dimensions (in our case, d-dimensional, where $d \geq C$) by learning linear combinations of the input channels. However, research on ShuffleNet Ma et al. (2018) and Tishby & Zaslavsky (2015); Han et al. (2021) has shown that extensive use of pointwise convolutions can decrease execution speed and potentially reduce accuracy if the expansion is too aggressive or misapplied. To address this, we propose improving the classic inverted bottleneck by making the first 1x1 convolution optional (if the dimension d is equal to the input dimension C). We define the input and output dimensions as C and the expanded dimension as d. For filters $W_1 \in \mathbb{R}^{1,1,C,d}$, $W_2 \in \mathbb{R}^{k,k,1,d}$, and $W_3 \in \mathbb{R}^{1,1,d,C}$, our approach can be represented as follows:

$$y = \begin{cases} W_3 \otimes [W_2 \otimes (W_1 \otimes x)] & \text{if } d \neq C \\ W_3 \otimes [W_2 \otimes (W_1 \otimes x)] & \text{if } d = C \text{ and } W_1 = \text{True} \\ W_3 \otimes [W_2 \otimes (x)] & \text{if } d = C \text{ and } W_1 = \text{False} \end{cases} \tag{1}$$

Similar to the state-of-the-art neural network techniques for object detection Wang et al. (2023; 2024), we consistently implement the SiLU Elfwing et al. (2018) activation function throughout our model. Appendix A.3.1 compares state-of-the-art and LeYOLO architecture.

### 3.1.2 LEYOLO BACKBONE - STRIDE STRATEGY.

We define each layer number of input and output channels as $C_i$ and the expansion dimension of the inverted bottleneck as $d_i$, with $d \geq C$ for each level of semantic information from $P_0 = 0$ - the model's input image - to $P_5 = 5$, the final spatial size after all reductions, with $P = [0, 1, 2, 3, 4, 5]$. For instance, $C_1$ and $d_1$ represent the number of channels corresponding to the spatial size after **one** downsampling convolution. We aim to enrich the information flow from the hidden layer defined as $h_j$ at the semantic information level $P_i$ to the subsequent hidden layer $h_{j+1}$ as $P_{i+1}$ by increasing the channels $C_i$ proportionately to the anticipated channel expansion from $d_{P_{i+1}}$.

**SiLU inverted bottleneck**

$$d = 3 \times C_i \qquad h'_j = SILU(bn(W_1 \otimes x + b)) \qquad W_1 \in \mathbb{R}^{1,1,C_i,d}$$

$$C = [16, 32, 64, 96] \qquad h''_j = SILU(bn(W_2 \otimes h'_j + b)) \qquad W_2 \in \mathbb{R}^{k,k,1,d}$$

$$x = h_j \qquad h_{j+1} = bn(W_3 \otimes h''_j + b) \qquad W_3 \in \mathbb{R}^{1,1,d,C_{i+1}}$$

**SiLU inverted bottleneck with stride (ours)**

$$d = [16, 16, 96, 192, 512] \qquad h'_j = SILU(bn(W_1 \otimes x + b))^3 \qquad W_1 \in \mathbb{R}^{1,1,C_i,d_{i+1}}$$

$$C = [16, 32, 64, 96] \qquad h''_j = SILU(bn(W_2 \otimes h'_j + b)) \qquad W_2 \in \mathbb{R}^{k,k,1,d_{i+1}}$$

$$x = h_j \qquad h_{j+1} = bn(W_3 \otimes h''_j + b) \qquad W_3 \in \mathbb{R}^{1,1,d_{i+1},C_{i+1}}$$

We find further channel expansion at downsampling levels at $P3$ and $P5$ in the LeYOLO backbone.

### 3.1.3 RELATIONSHIP TO DIMENSION CHOICE.

The information bottleneck principle theory from Tishby & Zaslavsky (2015) highlights two critical aspects of learning theory concerning information. Firstly, the authors recognize that deep neural networks (DNNs)

---

[3]Optional if $C_i = d_{i+1}$

are Markov Chains Gagniuc (2017) as $X \to \tilde{X} \to Y$ with $X$, $\tilde{X}$ and $Y$ being the input ($X$), the minimal sufficient statistics extracted from X ($\tilde{X}$) and the output ($Y$) respectively with $I(X; \tilde{X}) \geq I(\tilde{X}; Y)$. Consequently, to derive $\tilde{X}$ as the minimal sufficient statistics for extracting meaningful features to address $Y$, DNNs need to learn how to extract features using minimal sufficient statistics, employing the **most compact architecture** possible Tishby & Zaslavsky (2015).

Secondly, because DNNs only process inputs from the preceding layer $h_{i-1}$, a direct implication involves potentially losing information that subsequent layers cannot regain (equation (2)).

$$I(Y; X) \geq I(Y; h_i) \geq I(Y; h_{i+j}) \quad with \quad i + j \geq i \tag{2}$$

Expensive solutions such as column-oriented networks Cai et al. (2023); Hinton (2023) with intensive feature sharing between each block address this issue by incorporating intensive training blocks or adding additional detection heads at crucial points of information segmentation, as seen very recently in YOLOv9 Wang et al. (2024). As achieving equity in the equation above is feasible, the theory Tishby & Zaslavsky (2015) suggests that each layer should maximize information within itself $I(Y; h_i)$ while minimizing inter-layer information exchange as much as possible $h_i \to h_{i+1}$. Hence, rather than augmenting computational complexity in our model like Wang et al. (2023; 2024); Hinton (2023); Cai et al. (2023), we opted to scale it more efficiently, integrating Dangyoon Han's at al. Han et al. (2021) inverted bottleneck theory which stated that pointwise convolutions should not overpass a ratio of 6 in inverted bottleneck.

Our implementation involves minimizing inter-layer information exchange in the form of $I(X; h_1) \geq I(X; h_2) \geq ... \geq I(X; h_n)$, with $n$ equal to the last hidden layer of the neural network **backbone**, by ensuring that the number of input/output channels never exceeds a difference ratio of 6 from the first hidden layer through to the last. We define $h_i$ all the neurons from one inverted bottleneck as a whole (pointwise and depthwise convolutions). Therefore, $C$ from hidden layer $h_n$ is only 6 times greater than $C$ from hidden layer $h_1$. In this manner, we minimize inter-layer information exchange $h_i \to h_{i+1}$ as $h_1 \to h_n$. We maximize $I(Y; h_i)$ with an expansion of 3 in the whole network inverted bottlenecks, which correspond to an expansion of 6 from $P1 \to P4$ and 9 from $P4 \to P5$ regarding equations of our SiLU inverted bottleneck with stride in chapter 3.1.2.

More information on chapter experimental details A.5.

### 3.1.4 LeYOLO AS A GENERAL-PURPOSE OBJECT DETECTOR

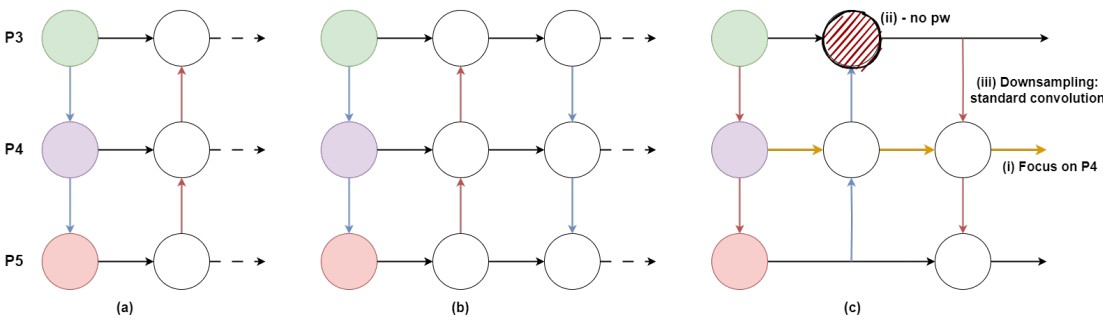

Figure 3: Difference between proposed LeYOLO neck as an efficient semantic feature aggregator. (a) Correspond to FPN Ghiasi et al. (2019). (b) Represent PANnet Zhao et al. (2017). Finally, (c) is our proposed solution.

**Neck.** In object detection, we call the neck the part of the model that aggregates several levels of semantic information, sharing extraction levels from more distant layers to the first layers. Historically, researchers have used a PANet Zhao et al. (2017) or FPN Lin et al. (2017) to share feature maps efficiently, enabling multiple detection levels by linking several semantic information $P_i$ to the PANet and their respective outputs as depicted in Figure 3(a).

**Figure 3.(i) Semantic information aggregation.** In this paper, we are mainly focusing on two competitors: BiFPN Tan et al. (2020) and YOLOF's SiSO Chen et al. (2021). BiFPN shares our model's central philosophy: using layers with low computational cost (concatenation and additions, depthwise and pointwise convolutions). However, BiFPN requires too much semantic information and too many blocking states (waiting for previous layers, complex graphs), which makes it difficult to keep up with fast execution speed. SiSO Chen et al. (2021), on the other hand, is interesting in its approach to object detection. Indeed, we can see that the authors of YOLOF have decided to use a single input and output for the model neck. Compared with other proposed solutions in YOLOF paper, we observe a significant degradation between a neck with multiple outputs (Single-in, Multiple-out - SiMO) and a neck with a single output (Single-in, Single-out - SiSO). We are particularly interested in their work on the potential efficiency of a SiMO, proving the possibility of improving the first layers of the neck of a YOLO model by optimizing the flow of semantic information with only one rich input.

We have identified a very important aspect in the composition of deep neural networks. During preliminary research on blocks specifically designed to reduce parameter count and FLOPs, we observed a recurring pattern in deep neural networks. Similar to the number of channels, it is difficult to determine the usefulness of a specific number of layer repetitions. However, we noticed that there is consistently a significant repetition of layers at the semantic level equivalent to P4. We found this in all MobileNets Howard et al. (2017); Sandler et al. (2018); Howard et al. (2019), in the optimization of inverted bottlenecks in EfficientNets Tan & Le (2019; 2021) and EfficientDet Tan et al. (2020), as well as in more recent architectures with self-attention mechanisms like MobileViTs Mehta & Rastegari (2022a;b), Wadekar & Chaurasia (2022), EdgeNext Maaz et al. (2023), and FastViT Vasu et al. (2023), which are designed for speed. Even more interestingly, models designed by NAS Tan et al. (2019); Howard et al. (2019); Tan & Le (2019) also utilize this pattern. Therefore, we support our supposition that P4 is the core of LeYOLO's neck. The backbone presented in the previous chapter uses a more intensive repetition of layers at the P4 semantic level.

**Figure 3.(ii) Efficient computation.** We reduce the computation - especially at P3 level because of the high spatial size - by removing the first pointwise convolution. After an ablation study performed on the LeYOLO nano-scaled backbone, we took the opportunity to remove time-costly pointwise convolutions since the input channels from the backbone P3 concatenated with the upsampled features from P4 results in the d-dimension required by the in-between depthwise convolution from our optimized inverted bottleneck presented in equation (1), chapter 3.1.1. Regarding information bottleneck theory, we minimized each neuron's interaction much further than the backbone. Each number of input channels, but also the number of expanded channels from the inverted bottleneck never exceeds 6. Input from P3 is $32C$ while the very last hidden layer of the LeYOLOs neck expanded channels $d$ equals 192.

**Figure 3.(iii) Standard strided convolution.** We improve the accuracy by using careful attention to stride details. As standard convolutions are not very parameters and computationally friendly, we thought of a way, in-bound with our low number of channels, and in regards to computation with stridden standard convolutions, to use them two times. From $P3$ to $P4$, and from $P4$ to $P5$. The gain of accuracy from such a choice proves its efficacity, regarding LeYOLO as a whole boost performance but also how LeYOLO *outperforms* SSDLite both in terms of parameters and precision as we discover in chapter 3.2.

**Decoupled Network in Network Head.** Until YOLOv5 Redmon et al. (2016); Redmon & Farhadi (2017; 2018); Bochkovskiy et al. (2020); Jocher et al. (2022), we had single model heads for classification and object detection. However, since YOLOv6 Li et al. (2022), the model head has become a more powerful tool, separating the block into two parts: *a branch for classification and object regression.* Although very efficient, this implies an almost doubled cost, requiring convolutions for classification and detection.

We theorize that there is no need to add spatial information other than to refine the features extracted by the backbone and neck channels-by-channels using lightweight depthwise convolutions (Figure 4). Through YOLO's point-by-point grid operation, we theorize that it is possible to simplify detection heads using pointwise convolution as a sliding multi-layer perceptron solution pixel by pixel, resembling classification propositions for each pixel. Several depthwise convolutions for spatial-only instructions refine the spatial relationship between two pointwise classifiers and regress each pixel.

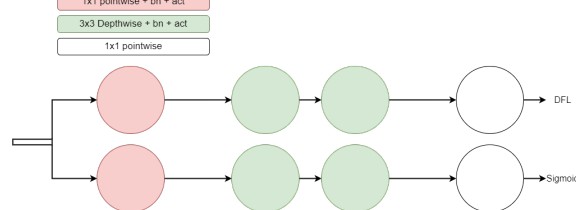

Figure 4: LeYOLO Head architecture

We prove that using only pointwise convolutions at the head of the model yields impressive results with 33.4 mAP at the LeYOLO-Nano@640 scale. Refining spatial information with depthwise convolution between pointwise convolutions pushes the model to **34.3 mAP**.

## 3.2 EXPERIMENTAL RESULTS

We train each neural network with the same exact hyperparameters and data augmentation, such as SGD, with a learning rate of 0.01 and momentum of 0.9. We mostly rely on mosaic data augmentation as well as hsv of $\{0.015, 0.7, 0.4\}$ and an image translation of $0.1$. As for the training specificities, we used a 96-batch size over 4 P100 GPUs. Performance is evaluated on the validation set using mean average precision. For more hyperparameters, see the appendix.

For LeYOLO, we offer a variety of models inspired by the architectural base presented above. A classic approach involves scaling the number of channels, layers, and input image size. Traditionally, scaling emphasizes channel and layer configurations, sometimes incorporating various scaling patterns.

LeYOLO scale from Nano to Large version with scaling related to what EfficientDet brought: $channels$ from $1.0$ to $1.33$, $layers$ from $1.0$ to $1.33$, and spatial size for training purpose from $640 \times 640$ to $768 \times 768$. Several spatial sizes are used for evaluation purposes, ranging from $320 \times 320$ to $768 \times 768$. Further information on scaling is in the appendix.

### 3.2.1 MOBILE OBJECT DETECTION

**Computation.** LeYOLO outperforms the state-of-the-art YOLO-type object detectors on embedded devices or those with limited computational power. In the appendix, we provide a detailed table showing the number of FLOPs, and we observe a correlation between this metric and the execution speed on low-computation devices, particularly in terms of parallelization. LeYOLO is faster than recent YOLO models designed for speed, achieving better accuracy (Table 1).

**Downstream tasks results.** We integrate several sota low-parameter backbones with LeYOLO neck and head, calling the resultant network the concerned backbone + LeYOLO. No matter the backbone used, all channel numbers, $P3, P4$ and $P5$

Table 1: LeYOLO speed (ms - lower is better) and accuracy ratio on embedded devices (Onnx GPU runtime - no trt).

| Models | Input Size | mAP | Speed(ms) |
|---|---|---|---|
| LeYOLO Nano | 320 | **25.2** | **44.7** |
| LeYOLO Nano | 640 | **34.3** | **74.0** |
| LeYOLO Small | 640 | **38.2** | **87.9** |
| LeYOLO Medium | 640 | **39.3** | **106.6** |
| YOLOv8 Nano Jocher et al. (2023) | 640 | 37.3 | 111.9 |
| YOLOv10 Nano cite | 640 | 38.5 | 126.6 |
| EfficientDet D0 Tan et al. (2020) | 518 | 34.6 | 137.0 |
| YOLOv9 Tiny Wang et al. (2024) | 640 | 38.3 | 144.9 |
| LeYOLO Large | 767 | 41.0 | 145.2 |
| EfficientDet D1 Tan et al. (2020) | 640 | 40.5 | 271.57 |

repetition specificities stay the same. At $P3$, the first pointwise convolution is never used, like in vanilla LeYOLO, resulting in the first filter being the depthwise convolution of the exact size of the backbone

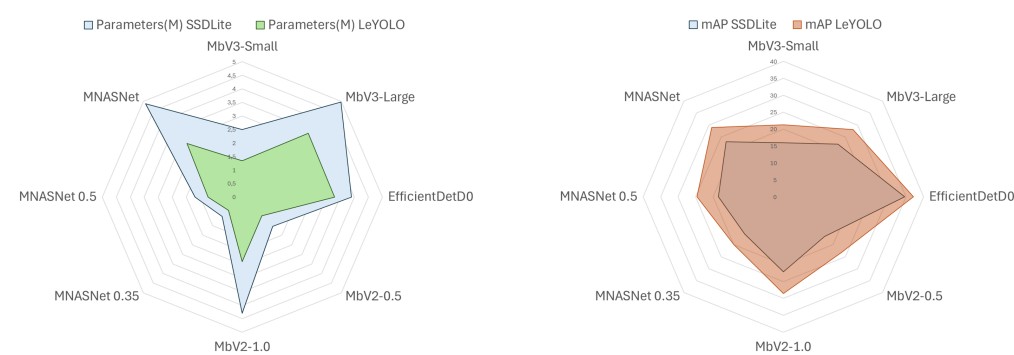

Figure 5: LeYOLO compared to SSDLite, with better parameter and precision efficiency

equivalent input number of channels.

*We take inspiration from SSDLite auxiliary parts:*

LeYOLO as a downstream object detector for lightweight classifiers keeps at the same number of channels [4] and layer repetition [5] as the LeYOLO Nano version. From a variety of lightweight classifiers with a low number of parameters and FLOP, LeYOLO outperformed SSDLite in every aspect of what we expect from a low-cost model - better parameter scaling, better precision, and finally, better throughput [6] - with results described in table 2 and Figure 5.

| Models | SSDLite | **LeYOLO** | SSDLite | **LeYOLO** |
|---|---|---|---|---|
| | Parameters(M) | | mAP.95 | |
| V3-Small | 2.49 | **1.34** | 16.0 | **21.3** |
| V3-Large | 4.97 | **3.33** | 22 | **28.1** |
| EfficientDetD0 | 3.9 | **3.29** | 34.6 | **37.1** |
| V2-0.5 | 1.54 | **0.98** | 16.6 | **23.3** |
| V2-1.0 | 4.3 | **2.39** | 22.1 | **28.6** |
| MNASNet 0.35 | 1.02 | **0.7** | 15.6 | **20.0** |
| MNASNet 0.5 | 1.68 | **1.22** | 18.5 | **24.6** |
| MNASNet | 4.68 | **2.8** | 23 | **28.9** |

Table 2: LeYOLO performance compared to lightweight classifier on MSCOCO object detection downstream tasks with SSDLite

# 4 CONCLUSION

As we try to offer thorough theoretical insights from state-of-the-art neural networks to craft optimized solutions, we acknowledge several areas for potential improvement, and we cannot wait to see further research advancements with **LeYOLO**.

---

[4]32 to 96 channels with extension ratio of 2

[5]repetition $l = 3$

[6]From the variety of available, fully reproducible and testable model on Jetsons devices with corresponding most up-to-data Jetpack versions

### 4.1 DICUSSIONS AND LIMITATIONS

**LeYOLO FPANet + DNiN Head:** Considering the cost-effectiveness of our FPANet and model head, there is a significant opportunity for experimentation across different backbones of state-of-the-art classification models. LeYOLO emerges as a promising alternative to SSD and SSDLite. The promising results achieved on MSCoco with our solution suggest potential applicability to other classification-oriented models We focused our optimization efforts specifically on **MSCOCO** and YOLO-oriented networks. However, we encourage experimentation with our solution on other datasets as well.

**Computational efficiency:** We have implemented a new scaling for YOLO models, proving that it is possible to achieve very high levels of accuracy while using very few computational resources (FLOP). Nevertheless, we are not the fastest in state of the art, as there are speed imperfections due to the (deliberate) lack of parallelizable architecture like our predecessors Sandler et al. (2018); Howard et al. (2019); Tan et al. (2020). However, only YOLOv7 and v6 are faster than our solution on a powerful enough GPU to ensure enough memory space on every YOLO benchmarked. As for the Jetson TX2, it seems LeYOLO is better blabla-finir. We could further analyze scaling for different edge powers to propose parallelizable column and block scaling.

### 4.2 FUTURE WORKS

We encourage further experimentation with our proposal, going deeper into experimental outcomes while exploring various dataset variants tailored to specific industry needs, such as intelligent agriculture and medicine.
We aim to provide a broader range of comparisons for LeYOLO in scenarios involving mobile devices with very limited computational resources, thereby demonstrating the ability of LeYOLO's low-memory-cost Neck to compute different levels of centralized semantic information on P4. Finally, as discussed in the limitations, LeYOLO could occupy a niche between parameter optimization and high execution speed, further advancing current object detection solutions for embedded systems.

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

# A  APPENDIX

## CONTENTS

## A.1 COMPLETE STATE-OF-THE-ART

In the paper, we presented comparisons based on criteria specific to each class of object detectors: speed for the YOLO mainline (Table X) and parameter count for low-cost detectors (Table Z). In this section, we expand the discussion by comparing the number of FLOPs. Additionally, we have enhanced the analysis with a more comprehensive comparative study, incorporating different levels of precision measurements. This section comprehensively compares the state-of-the-art, juxtaposing LeYOLO with YOLO mainline models, micro neural networks designed for object detection, and the leading classification model, SSDLite, operating at 320x320 resolution.

We evaluate our performance against others using two primary metrics: mean average precision (mAP) and FLOPs. The mAP is computed with various parameters, including an IOU of 0.5. For FLOP, somehow the intertwined nature of MAC and FLOP formulas has resulted in inconsistencies, with researchers erroneously labeling their models using FLOP instead of MAC. This implies that the computational cost of the model is at least twice the stated initial amount.

We also added the number of parameters used in a lightweight state-of-the-art model for object detection. Except for impressive results from YOLOv9-Tiny with two million parameters, our contributions use very few parameters compared to others. Table 3 shows all results.

Table 3: State-of-the-art lightweight object detector.

| Models | Input Size | mAP | mAP50 | mAP75 | S | M | L | FLOP(G) | Parameters (M) |
|---|---|---|---|---|---|---|---|---|---|
| MobileNetv3-SHoward et al. (2017) | 320 | 16.1 | - | - | - | - | - | **0.32** | 1.77 |
| MobileNetv2-x0.5Sandler et al. (2018) | 320 | 16.6 | - | - | - | - | - | 0.54 | 1.54 |
| MnasNet-x0.5Tan et al. (2019) | 320 | 18.5 | - | - | - | - | - | 0.58 | 1.68 |
| **LeYOLO-Nano** | 320 | **25.2** | **37.7** | **26.4** | **5.5** | **23.7** | **48.0** | 0.66 | **1.1** |
| MobileNetv3Howard et al. (2019) | 320 | 22 | - | - | - | - | - | 1.02 | 3.22 |
| YOLOX-NanoGe et al. (2021) | 320 | 25.3 | - | - | - | - | - | 1.08 | **0.91** |
| NanoDet | | 23.5 | - | - | - | - | - | 1.2 | 0.95 |
| **LeYOLO-Small** | 320 | **29** | 42.9 | 30.6 | 6.5 | 29.1 | 53.4 | 1.126 | 1.9 |
| **LeYOLO-Nano** | 480 | **31.3** | **46** | **33.2** | **10.5** | **33.1** | **52.7** | 1.47 | 1.1 |
| MobileNetv2Sandler et al. (2018) | 320 | 22.1 | - | - | - | - | - | 1.6 | 4.3 |
| MnasNetTan et al. (2019) | 320 | 23 | - | - | - | - | - | 1.68 | 4.8 |
| **LeYOLO-Small** | 480 | **35.2** | 50.5 | 37.5 | 13.3 | 38.1 | 55.7 | **2.53** | **1.9** |
| Tinier-YOLO | | 17 | 34 | 15.7 | 4.8 | 17.3 | 26.8 | 2.563 | - |
| MobileNetv1Howard et al. (2017) | 320 | 22.2 | - | - | - | - | - | 2.6 | 5.1 |
| **LeYOLO-Medium** | 480 | **36.4** | **52.0** | **38.9** | 14.3 | 40.1 | 58.1 | 3.27 | 2.4 |
| **LeYOLO-Small** | 640 | **38.2** | **54.1** | **41.3** | **17.6** | **42.2** | 55.1 | **4.5** | **1.9** |
| YOLOv5-nJocher et al. (2022) | 640 | 28 | 45.7 | - | - | - | - | 4.5 | 1.9 |
| EfficientDet-D0Tan et al. (2020) | 512 | 33.80 | 52.2 | 35.8 | 12 | 38.3 | 51.2 | 5 | 3.9 |
| **LeYOLO-Medium** | 640 | **39.3** | **55.7** | **42.5** | **18.8** | **44.1** | **56.1** | 5.8 | **1.9** |
| YOLOv7-TinyWang et al. (2023) | 416 | 33.3 | 49.9 | - | - | - | - | 5.8 | 6.2 |
| YOLOX-TinyGe et al. (2021) | 416 | 32.8 | 50.3 | - | - | - | - | 6.5 | 5.06 |
| YOLOv4-tinyBochkovskiy et al. (2020) | - | 21.7 | - | - | - | - | - | 6.96 | 6.06 |
| YOLOv9-TinyWang et al. (2024) | 640 | 38.3 | 53.1 | 41.3 | - | - | - | 7.7 | 2 |
| **LeYOLO-Large** | 768 | **41** | **57.9** | **44.3** | **21.9** | **46.1** | **56.8** | **8.4** | **2.4** |
| YOLOv6-NLi et al. (2022) | 640 | 35.9 | 51.2 | - | - | - | - | 11.1 | 4.3 |

## A.2 NOTATIONS

Throughout the document, we use several notations to describe the essential components of deep learning, particularly in object detectionfor example, the spatial size of different tensors described as $P_i$. This chapter covers all the notations used in the paper. Since there is little to no consensus on Deep Learning notations, we found it relevant to describe them in more depth in the appendix for readers who need further explanation.

To start, we want to provide more explanation on the main component of this paper: the computation formula. We consistently use the **FLOP** (Floating Point Operations) metric to compare our work with other state-of-the-art neural networks throughout the paper. By basing computations on the number of multiplications and additions required for the neural network, we establish a solid foundation for efficient model comparisons. The FLOP metric remains reliable regardless of the hardware used, making it a good indicator of computational efficiency. Although we could use other metrics such as speed, these are highly dependent on factors like neural network architecture parallelization, the hardware used, the accelerator software (TensorRT, CoreML, TFLite), and memory usage and transfer speeds.

The second main element we consistently use throughout the paper is **mAP** (mean Average Precision). Researchers widely use this metric to compare object detection-based neural networks. mAP measures the precision of the model by evaluating the overlap between the proposed bounding boxes and the actual annotated bounding boxes. While some papers compare models using mAP at a fixed threshold of 50% overlap (mAP50), we primarily use mAP50-95. This metric averages the precision over different overlap thresholds, ranging from 50% to 95%, covering a broader range of evaluation criteria.

In object detection, where the spatial size of the feature map is essential, we define $P_i$ as the feature map size of our deep neural network. The sizes range from $P0$ (640x640 pixels) to $P5$ (20x20 pixels) for LeYOLO-Small to Medium, with $i$ representing the number of strides used. Similarly, $P_{i-1}$ denotes the size of the preceding feature map for the explicitly described feature map $i$.

Similarly, when describing hidden layers in the neural network, we refer to the entire block rather than a single convolution. For example, the paper describes a single inverted bottleneck as one hidden layer $h_j$ that consists of two pointwise convolutions and one depthwise convolution. Throughout the paper, this lets us directly refer to the preceding inverted bottleneck as $h_{j-1}$.

### A.3 EXTENDED DISCUSSION ON RELATED WORK

Sandler et al. (2018); Howard et al. (2019): Inspired by MobileNetv2's achievements in reducing the number of parameters and FLOPs in neural networks through inverted bottlenecks, MobileNetv3 demonstrated once again that it is possible to further leverage this architecture, which dates back to 2018, to significantly improve accuracy. This idea led to the development of LeYOLO, an optimization of the inverted bottleneck focused on finding the optimal number and arrangement of layers based on the spatial size of the inputs.
We advanced the concept of a more efficient object detection model by first implementing MobileNetv3-Small within the YOLOv8 API Jocher et al. (2023), yielding much more promising results than those achieved with SSDLite. However, we found the MobileNetv3-Small backbone too slow to reach the extremely fast execution speeds desired in YOLO-related research.

Tishby & Zaslavsky (2015); Han et al. (2021): Inspired by the study by Tishy et al., we compared the layer count of MobileNetv3 with a more reduced and consequently faster configuration. This led us logically to analyze the work of Dooyan et al. Through a deeper examination of layer count to understand its impact, they concluded that reducing the number of expansion layers rather than extending them, regardless of scaling is more beneficial. We extended this study by demonstrating that it can improve accuracy with expansions smaller than 3 throughout the neural network.
In LeYOLO, each part of the model never exceeds an input and output layer count difference of more than 6. The backbone of LeYOLO starts with 16 layers at P1 and ends with 96 layers. The Neck of LeYOLO has a layer count between 32 and 96, with a maximum expansion of 192, ensuring that the layer difference never exceeds 6. Finally, LeYOLOs Head is proportional to the number of input layers and MSCOCO classes.

Nakkiran et al. (2021): Nakkiran et al. demonstrated a new way of observing the scaling of deep neural networks, primarily addressing why many parameters on very deep neural networks work well for solving computer vision problems. Their study shows that there is indeed a model scaling that can enhance accuracy

with a very large number of parameters. However, they also observed a phenomenon of double descent when the training curve is plotted not against the number of epochs on the x-axis but against scaling. A regime with few parameters can outperform much larger scalings in this scenario. Therefore, this case study suggests that focusing on a very small number of parameters and performing scaling that fits this range rather than trying to surpass this barrier is worthwhile.

As a result, LeYOLO proposes a simple yet controlled scaling approach with a small amplifier for the number of layers and layer repetitions. The final scaling proposed is not based on the number of parameters in the neural network but on its input to maintain the integrity of a modest scaling approach.

Howard et al. (2017): MobileNetv4, which came out in the same period we initially worked on LeYOLO, proposes a similar alternative to the inverted bottleneck as we did. They propose to add an **optional** convolution in the inverted bottleneck: a new optional depthwise convolution right at the beginning. We are very interested in the study of MobileNetv4 as they focus on neural architecture search with speed rewards on mobile devices, and we raised the question, *does focusing on mobile hardware speeds reduce the number of parameters and FLOP of the neural network?*
MobileNetv4 is very fast but uses many more parameters than previously done in the state-of-the-art lightweight neural network but paved a very interesting way into shaping a more speed-efficient inverted bottleneck.

We consider LeYOLO and MobileNetv4 complementary as we prove that inverted bottleneck might still be more parameters-efficient, reaching better accuracy with a much smaller number of parameters and computation cost.

### A.3.1 ARCHITECTURE DIFFERENCES

We propose a comparison between our proposition and several inverted bottleneck-inspired backbones. Despite being disregarded by most contemporary state-of-the-art object detectors, MobileNetv2 Sandler et al. (2018), MobileNetv3 Howard et al. (2019), and EfficientNets Wang et al. (2020); Tan & Le (2021); Tan et al. (2020) share the philosophy of employing inverted bottlenecks for object detection.

As mentioned in the introduction, advancements in GPUs have enabled the development of powerful and rapid neural networks. However, inverted bottlenecks provide limited depth for parallelizing multiple computation blocks. Parallelizing deep neural networks on embedded devices remains challenging, but there is optimism for the future. Research primarily focuses on reducing MAC and FLOP costs and occasionally even memory access costs. Naturally, execution speed remains a significant concern. Nevertheless, we aim to briefly compare our backbone (Table 4) using consistent notation with models that exhibit "similar" architecture (Tables 6, 5 and 7), particularly those utilizing inverted bottlenecks.

Through this comparison, we can observe the stride-inverted bottleneck strategy. Upon code verification, we indeed notice a contrast in channel expansion when transitioning from one layer $(h_i; P_i)$ to another $(h_{i+1}; P_{i+1})$ with a stride greater than one. Additionally, most inverted bottlenecks utilize an expansion ratio of 6, whereas we only expand to 3 within a block. This reduces overall computation and allows the inverted bottleneck stride strategy to expand the number of channels one last time before the stride within the depthwise convolution.

Table 4: LeYOLO backbone (base scale: nano).

| Input | Operator | exp size | out size | NL | s |
|---|---|---|---|---|---|
| P0 | conv2d, 3x3 | - | 16 | SI | 2 |
| P1 | conv2d, 1x1 | 16 | 16 | SI | 1 |
| P1 | bneck, 3x3, **pw=False** | 16 | 16 | SI | 2 |
| P2 | bneck, 3x3 | **96** | 32 | SI | 2 |
| P3 | bneck, 3x3 | 96 | 32 | SI | 1 |
| P3 | bneck, 5x5 | 96 | 64 | SI | 2 |
| P4 | bneck, 5x5 | 192 | 64 | SI | 1 |
| P4 | bneck, 5x5 | 192 | 64 | SI | 1 |
| P4 | bneck, 5x5 | 192 | 64 | SI | 1 |
| P4 | bneck, 5x5 | 192 | 64 | SI | 1 |
| P4 | bneck, 5x5 | **576** | 96 | SI | 2 |
| P5 | bneck, 5x5 | 576 | 96 | SI | 1 |
| P5 | bneck, 5x5 | 576 | 96 | SI | 1 |
| P5 | bneck, 5x5 | 576 | 96 | SI | 1 |
| P5 | bneck, 5x5 | 576 | 96 | SI | 1 |

Table 5: MobileNetv3 backbone.

| Input | Operator | exp size | out size | NL | s |
|---|---|---|---|---|---|
| P0 | conv2d, 3x3 | - | 16 | HS | 2 |
| P1 | conv2d, 1x1 | 16 | 16 | RE | 2 |
| P2 | bneck, 3x3 | 72 | 24 | RE | 2 |
| P3 | bneck, 3x3 | 88 | 24 | RE | 1 |
| P3 | bneck, 5x5 | 96 | 40 | HS | 2 |
| P4 | bneck, 5x5 | 96 | 40 | HS | 1 |
| P4 | bneck, 5x5 | 240 | 40 | HS | 1 |
| P4 | bneck, 5x5 | 240 | 40 | HS | 1 |
| P4 | bneck, 5x5 | 120 | 48 | HS | 1 |
| P4 | bneck, 5x5 | 144 | 48 | HS | 1 |
| P4 | bneck, 5x5 | 288 | 96 | HS | 2 |
| P5 | bneck, 5x5 | 576 | 96 | HS | 1 |
| P5 | bneck, 5x5 | 576 | 96 | HS | 1 |

Table 6: MobileNetv2 backbone.

| Input | Operator | exp size | out size | NL | s |
|---|---|---|---|---|---|
| P0 | conv2d, 3x3 | - | 32 | RE | 2 |
| P1 | bneck, 3x3 | 16 | 16 | RE | 1 |
| P1 | bneck, 3x3 | 96 | 24 | RE | 2 |
| P2 | bneck, 3x3 | 144 | 24 | RE | 1 |
| P2 | bneck, 3x3 | 144 | 32 | RE | 2 |
| P3 | bneck, 3x3 | 192 | 32 | RE | 1 |
| P3 | bneck, 3x3 | 192 | 32 | RE | 1 |
| P3 | bneck, 3x3 | 192 | 64 | RE | 2 |
| P4 | bneck, 3x3 | 384 | 64 | RE | 1 |
| P4 | bneck, 3x3 | 384 | 64 | RE | 1 |
| P4 | bneck, 3x3 | 384 | 64 | RE | 1 |
| P4 | bneck, 3x3 | 384 | 96 | RE | 1 |
| P4 | bneck, 3x3 | 576 | 96 | RE | 1 |
| P4 | bneck, 3x3 | 576 | 96 | RE | 1 |
| P4 | bneck, 3x3 | 576 | 160 | RE | 2 |
| P5 | bneck, 3x3 | 960 | 160 | RE | 1 |
| P5 | bneck, 3x3 | 960 | 320 | RE | 1 |
| P5 | conv, 1x1 | 1920 | 320 | RE | 1 |

Table 7: EfficientNet-B0 (EfficientDet-D0 and D1) backbone.

| Input | Operator | exp size | out size | NL | s |
|---|---|---|---|---|---|
| P0 | conv2d, 3x3 | - | 32 | RE | 2 |
| P1 | bneck, 3x3 | 16 | 16 | RE | 1 |
| P1 | bneck, 3x3 | 96 | 24 | RE | 2 |
| P2 | bneck, 3x3 | 144 | 24 | RE | 1 |
| P2 | bneck, 5x5 | 144 | 40 | RE | 2 |
| P3 | bneck, 5x5 | 240 | 40 | RE | 1 |
| P3 | bneck, 3x3 | 240 | 80 | RE | 1 |
| P3 | bneck, 3x3 | 480 | 80 | RE | 1 |
| P3 | bneck, 3x3 | 480 | 80 | RE | 1 |
| P3 | bneck, 5x5 | 480 | 112 | RE | 2 |
| P4 | bneck, 5x5 | 672 | 112 | RE | 1 |
| P4 | bneck, 5x5 | 672 | 112 | RE | 1 |
| P4 | bneck, 5x5 | 672 | 192 | RE | 2 |
| P5 | bneck, 5x5 | 1152 | 192 | RE | 1 |
| P5 | bneck, 5x5 | 1152 | 192 | RE | 1 |
| P5 | bneck, 5x5 | 1152 | 192 | RE | 1 |
| P5 | bneck, 3x3 | 1280 | 320 | RE | 1 |

## A.4 OVERALL ARCHITECTURE

For the sake of readability, we have omitted a discussion of the semantic shcostg strategy in the neck. In the bottom-up pathway, we employ lightweight blocks for upsampling the feature map, while the top-down pathway utilizes standard convolutions. Despite their pronounced cost, strived standard convolutions prove efficient in this context due to their low spatial size and the restrained number of channels used.

We experimented with inverted bottlenecks instead of convolutions but found them more costly and less accurate. Additionally, we refrain from using any convolutions before the first 20x20 upsampling, diverging from the approach taken in most YOLO architectures. We question the necessity of another convolution after the Spatial Pyramid Pooling Fusion (SPPF) Jocher et al. (2022; 2023), as it may already be sufficiently efcostent in the backbone. Lastly, the 80x80 aspect is resource-intensive and requires careful consideration. Following similar reasoning, we avoided computation for the 80x80 top-down pathway, as the cost seemed disproportionate to the marginal accuracy improvement, as demonstrated in ablation studies on FPANet's 80x80 pointwise component. Figure 6 represents the complete architecture of LeYOLO.

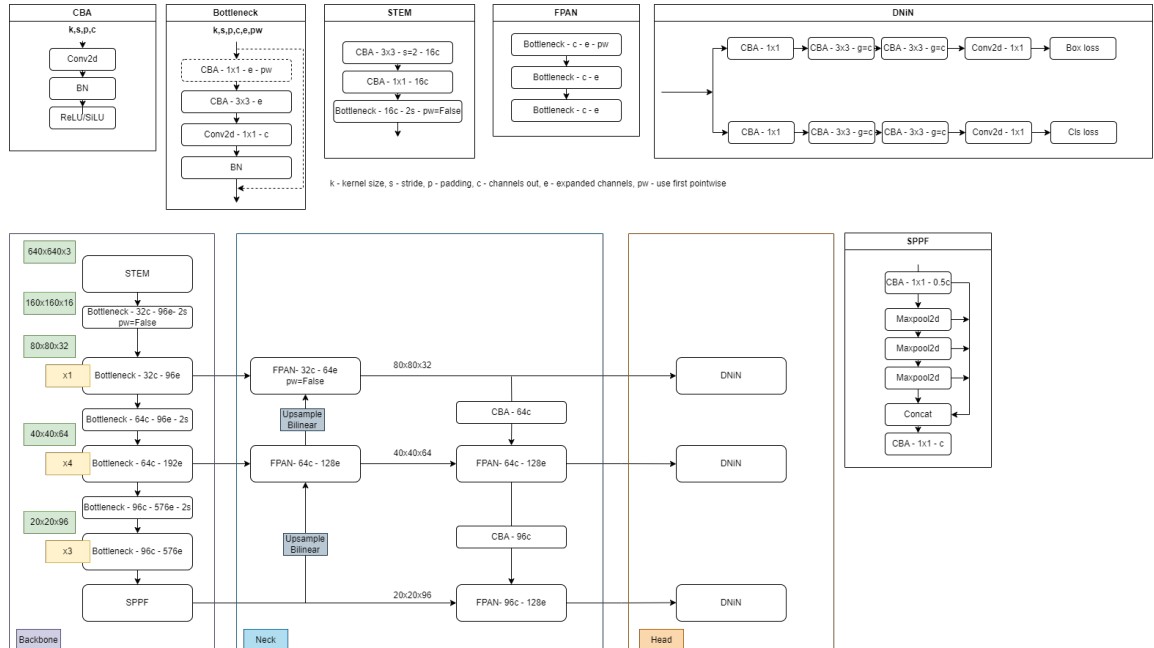

Figure 6: We present a visual decomposition of the LeYOLO network's comprehensive architecture, explaining its backbone, neck, and head components.

## A.5 EXPERIMENTAL DETAILS

### A.5.1 MODELS

To observe the effectiveness of our theories presented in the chapter on the information bottleneck with a relatively low parameter scaling, we propose using a family of architectures in the form of a "toy model," as seen in the "Deep Double Descent" paper Nakkiran et al. (2021), with an inverted bottleneck as the basic block.

**Inverted Bottleneck.** Our inverted bottleneck is structured similarly to the backbone of LeYOLO. It consists of three convolutions: two pointwise and one depthwise placed between them. The first two convolutions are followed by batch normalization and SiLU activation. The final pointwise convolution only uses normalization without activation.

**Architecture.** The overall architecture of the toy model begins with a standard $3x3$ convolution to a channel count of $k$, followed by a pointwise convolution that maintains the same number of channels, similar to the input processing in LeYOLO. The core of the backbone consists of **four** layers inspired by the work of Nakkiran et al. (2021). Each layer comprises two inverted bottlenecks, with the first one applying its corresponding stride and the number of channels expanding from $k$ to $k = [k, 2k, 4k, 8k]$, using strides of $s = [1, 2, 2, 2]$. The expansion factor in the inverted bottleneck is capped at 3, with $e = [3, 3, 3, 3]$.

We are studying the capacity of the information bottleneck, as described in Chapters 3.1.2 and 3.1.3, with $h_i, 0 \leq i < n$ where $n = 4$ refers to the four layers of the toy model.

### A.5.2 EXPERIMENTATIONS - IMAGE CLASSIFICATION

To validate our ideas on the information bottleneck, particularly regarding the optimization of layers $h_i$ for $I(Y; h_i)$ while minimizing the interconnection between layers from $h_i$ to $h_{i+1}$ from chapter 3.1.3 and chapter 3.1.2, we propose this experiment with $k$ varying from 16 to 64 as the starting point for our toy model on CIFAR10.

As indicated in the chapter on the information bottleneck, we minimize $h_i \rightarrow h_{i+1}$ in the form of $h_1 \rightarrow h_n$, and in our experiments, we adjust the scaling from $k = [k, 2k, 4k, 8k]$ to $k = [k, 2k, 4k, \mathbf{6}k]$. This aligns with the idea of not expanding the input and output information too much, maintaining a global difference of **6** or less.

We focus on increasing the channels in the inverted bottleneck to maximize $h_i$ as much as possible. We propose comparing the model's performance with an expansion ratio $e$ between $e = 3$ and $e = 6$.

Finally, we combine this study with our strategy to maximize $h_i$ in an inverted bottleneck with a stride. In the state-of-the-art approach, this is typically capped at the current layer's $k$ value, not the one targeted after the stride. We then implement a comparison using an expansion of $e = 6$ for strides $>= 2$, and $e = 3$ as the standard expansion for the other inverted bottlenecks. The solution we implemented for LeYOLO, which optimizes its information bottlenecks, also demonstrates its superiority on the CIFAR10 testbed, outperforming all other state-of-the-art approaches for channel expansion using inverted bottlenecks, as shown in Figure 7 (label: max x6 x3dw x6 stride).

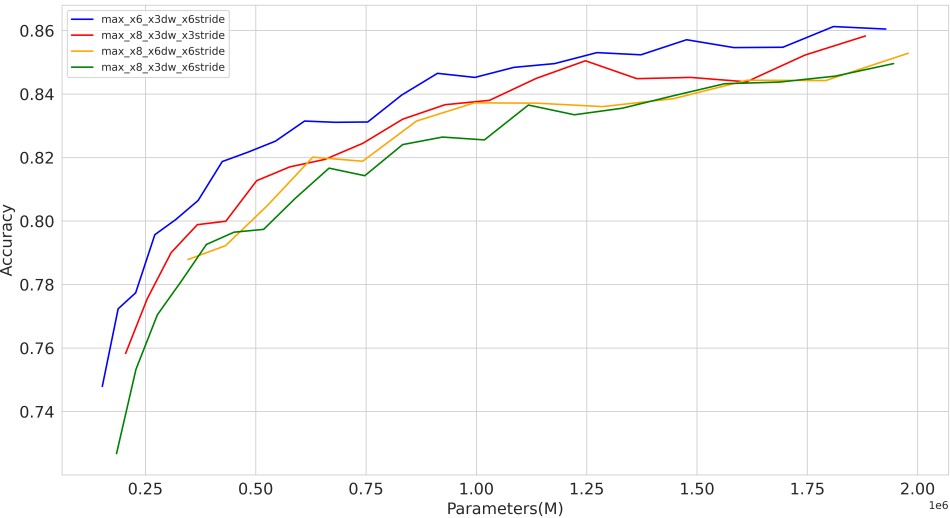

Figure 7: CIFAR10 Information bottleneck experimentation with inverted bottlenecks with $k$ varying from 16 to 64

**max x8 x3dw x3stride.** Base experimentation with $k = [k, 2k, 4k, 8k]$, $e = [3, 3, 3, 3]$, $s = [1, 2, 2, 2]$.
**max x8 x6dw x6stride.** Experimentation with $k = [k, 2k, 4k, 8k]$, $e = [6, 6, 6, 6]$, $s = [1, 2, 2, 2]$.
**max x8 x6dw x3stride.** Experimentation with $k = [k, 2k, 4k, 8k]$, $e = [6, 6, 6, 6]$ with strided convolution and $e = [3, 3, 3, 3]$ for non-strided ones, $s = [1, 2, 2, 2]$.
**max x6 x3dw x6stride.** Experimentation with $k = [k, 2k, 4k, \mathbf{6}k]$, $e = [6, 6, 6, 6]$ with strided convolution and $e = [3, 3, 3, 3]$ for non-strided ones, $s = [1, 2, 2, 2]$.

Table 8: LeYOLO base training scaling architecture with their respective results

| Models | Nano | Small | Medium | Large |
|---|---|---|---|---|
| Input spatial size | 640 | 640 | 640 | 768 |
| Channels ratio | x1 | x1.33 | x1.33 | x1.33 |
| Layer ratio | x1 | x1 | x1.33 | x1.33 |
| mAP | 34.3 | 38.2 | 39.3 | 41 |

### A.5.3 ARCHITECTURE SCALING CHOICE

This section provides more insight into the model scaling proposed in our contributions. As discussed in the chapter 3.2, we propose four different scaling methods, compressing the architecture below 10 FLOP(G). Table 8 illustrates the four training scaling possibilities (Nano to Large), and Table 9 shows the eight final proposed scaling for inferences.

Table 9: Architecture scaling with different parameter multipliers.

| Models | Nano | Nano | Small | Small | Small | Medium | Medium | Large |
|---|---|---|---|---|---|---|---|---|
| Input spatial size | 320p | 480p | 320p | 480p | 640p | 480p | 640p | 768p |
| Channels ratio | x1 | x1 | x1.33 | x1.33 | x1.33 | x1.33 | x1.33 | x1.33 |
| Layer ratio | x1 | x1 | x1 | x1 | x1 | x1.33 | x1.33 | x1.33 |

We can effectively transfer the core architecture from the previously mentioned ablation study to different input sizes during inference and validation. For instance, a neural network explicitly trained at 640p might yield better results when compressed and validated at 320p than a neural network trained from scratch at 320p. Consequently, we tested various scales of LeYOLONano, Small, Medium, and Large trained neural networks from Table 8 to determine the optimal input and training combinations. The results presented in Table 10 highlight the best outcomes from this study.

Table 10: Ablation study on best training and validation input size

| Models | training dim | validation dim | mAP | FLOP(G) |
|---|---|---|---|---|
| LeYOLO-Nano | 320 | 320 | 24.1 | 0.66 |
| **LeYOLO-Nano** | 640 | 320 | **25.2** | **0.66** |
| LeYOLO-Nano | 480 | 480 | 30.9 | 1.47 |
| **LeYOLO-Nano** | 640 | 480 | **31.3** | **1.47** |
| **LeYOLO-Nano** | 640 | 640 | 34.3 | 2.65 |
| **LeYOLO-Small** | 640 | 320 | **29.0** | **1.126** |
| **LeYOLO-Small** | 640 | 480 | **35.2** | **2.53** |
| **LeYOLO-Small** | 640 | 640 | **38.2** | **4.51** |
| LeYOLO-Medium | 640 | 320 | 30.0 | 1.45 |
| **LeYOLO-Medium** | 640 | 480 | **36.4** | **3.27** |
| **LeYOLO-Medium** | 640 | 640 | **39.3** | **5.8** |
| **LeYOLO-Large** | 768 | 768 | **41** | **8.4** |

### A.6 SPEED TESTS

As discussed in the paper, we proposed a highly efficient family of neural network models, focusing solely on FLOP computation and disregarding execution speed. Inverted bottlenecks inherently reduce the parallelization potential of the neural network, causing GPUs to wait for subsequent operations sequentially.

Table 11: Execution speed of reproducible YOLOs and our contribution (sorted by queries per second).

| Models | QPS | FLOP(G) | mAP(%) |
|---|---|---|---|
| **LeYOLO-Nano@320** | 99.56 | 0.66 | 25.2 |
| **LeYOLO-Small@320** | 75.36 | 1.126 | 29 |
| **LeYOLO-Nano@480** | 51.39 | 1.47 | 31.3 |
| **LeYOLO-Small@480** | 39.29 | 2.53 | 35.2 |
| YOLOv5n | 38 | 4.5 | 28 |
| YOLOv6n | 37.935 | 11.1 | 35.9 |
| YOLOv8n | 33.650 | 8.7 | 37.3 |
| **LeYOLO-Medium@480** | 32.83 | 3.27 | 36.4 |
| YOLOv7-Tiny | 24.8 | 5.8 | 33.3 |
| **LeYOLO-Small@640** | 24 | 4.5 | 38.2 |
| **LeYOLO-Medium@640** | 19.89 | 5.8 | 39.3 |
| YOLOX-s | 14.6 | 26.8 | 40.5 |
| **LeYOLO-Large@768** | 14.2 | 8.4 | 41 |

Consequently, while our models may not be the fastest in the state-of-the-art using TensorRT, they offer various models with varying execution speeds. We focus on object detectors on embedded devices, so we propose a comparison using a 4GB Jetson TX2 coupled with the TensorRT software accelerator to observe the state-of-the-art parallelization capability. We can find details of execution speed, accuracy, query per second, FLOP, and qps in Table 11. However, not all models are fully compatible with TensorRT accelerations, and most use special tricks to make it work; therefore, the mAP can't be solely verified. LeYOLO, on the other hand, is fully compatible with TensorRT; no further graph surgeon is necessary.

## A.7 CODE

As we could use PyTorch, Tensorflow, or any other API, we are using the Ultralytics code on the YOLOv8 version to develop our version of LeYOLO. Using these tools and implementing the code will be simple, centralizing research on a single tool.

## A.8 TRAINING SPECIFICITY

**Training on MSCOCO.** We train our model on the MSCOCO dataset Lin et al. (2014) using the standard data augmentation [49] with stochastic gradient descent (SGD) and batch size of 128 on four GPUs. Learning rate is initially set to 0.01 with a momentum set to 0.9. Weight decay is set to 0.001.

**Mosaic data augmentation :** throughout the training, we found through multiple experiments that there is minimal variation in accuracy attributable to Mosaic data augmentation. This phenomenon primarily arises from small objects with limited data samples, such as toothbrushes in MSCOCO, where Mosaic augmentation could potentially have adverse effects. Across our experiments, we noted a potential variance of 0.4 mAP.

1. epochs: 500

2. patience: 50

3. batch: 128

4. imgsz: 640

5. gpu count: 4

6. workers: 8

7. optimizer: SGD

8. seed: 0

9. close mosaic: 10

10. training iou: 0.7

11. max detectections: 300

12. lr0: 0.01

13. lrf: 0.01

14. momentum: 0.9

15. weight decay: 0.001

16. warmup epochs: 3.0

17. warmup momentum: 0.8

18. warmup bias lr: 0.1

19. box: 7.5

20. cls: 0.5

21. dfl: 1.5

22. pose: 12.0

23. kobj: 1.0

24. label smoothing: 0.0

25. nbs: 64

26. hsv h: 0.015

27. hsv s: 0.7

28. hsv v: 0.4

29. degrees: 0.0

30. translate: 0.1

31. scale: 0.5

32. shear: 0.0

33. perspective: 0.0

34. flipud: 0.0

35. fliplr: 0.5

36. mosaic: 1.0

37. mixup: 0.0

38. copy paste: 0.0

39. erasing: 0.4

40. crop fraction: 1.0

