# OpenReview forum: "LeYOLO: Lightweight, Scalable and Efficient CNN Architecture for Object Detection"
_ICLR.cc/2025/Conference — Submitted to ICLR 2025_

### Official Review · Reviewer_B89x · 2024-10-25

**Soundness:** 3
**Presentation:** 2
**Contribution:** 2
**Rating:** 3
**Confidence:** 5

**Summary:**

This paper aims to present an efficient network architecture for real-time object detection, called LeYOLO. The basic goal is to investigate a better way to design state-of-the-art fast object detectors based on lightweight classifiers, like MobileNets and EfficientNet. The authors rethink the design of basic blocks and the architecture of detectors, which take the inverted bottleneck structure as cores. Experiments show that the proposed object detectors works well on COCO object detection and performs better than previous state-of-the-art real-time object detectors.

**Strengths:**

- The starting point is interesting. Unlike previous real-time object detections, like the YOLO series, this paper borrows the techniques from previous popular efficient image classifiers to design the network architecture.
- The experimental results are good. Compared to previous popular real-time object detectors, the proposed approach receives better results on the COCO object detection benchmark.

**Weaknesses:**

- The presentation of this paper is not good.

-- In the related work section, the authors use a long paragraph to describe the advantages of the proposed method over previous work. This looks strange. I recommend the authors to move this paragraph to the method section. After all, the proposed method has not been introduced in the related work section.

-- It really takes me some time to understand the main technical details of the proposed approach. The method section is really difficult to understand. I recommend the authors to reoutline this section. Maybe, the main architecture can be put at the beginning of this section.

- The novelty of this paper should be made clearer. It is true that most previous real-time object detectors did not clearly explain that they borrows techniques from popular efficient image classifiers but many techniques from efficient image classifiers have been used in modern object detectors, like the inverted bottleneck structure. I do not think the proposed method is of great technical contributions.

- Another concern is considering that the authors borrow ideas from popular efficient image classifiers, why this is important from the design of object detectors? According to my knowledge, object detection and image classification are two different tasks. I think the authors should make this clear.

- Though the results on the COCO dataset are good, it seems that the authors did not provide ablation experiments to analyze why the proposed techniques work well. There are some in the appendix but these experiments are not about the network design.

- I also recommend the authors to re-draw some of the figures. Some text in the figures are really difficult to see clear, for example, the ones in Fig. 1. Why not make them larger?  In addition, some of the text are not clearly explained. For example, what are the meanings of the text in Fig. 3(c), like 'no pw'?

**Questions:**

- The technical contributions of this paper should be made clearer.

- The presentation of this paper should be largely improved.

- More ablation experiments should be added to explain how each component in the proposed approach works.

---

### Official Review · Reviewer_WgRi · 2024-10-25

**Soundness:** 2
**Presentation:** 1
**Contribution:** 2
**Rating:** 3
**Confidence:** 4

**Summary:**

This paper propose a lightweight architecture for YOLO-like object detectors. The proposed architecture is built on the classic inverted bottleneck structure. The paper proposed to make the first pointwise convolution optional when the input channel is the same as the expansion channel to reduce computational cost. The paper also decide the number of expansion channel in the inverted bottleneck differently compared to the original design. For the neck architecture, the paper proposed to put focus on P4. For head architecture, the paper proposed to use lightweight operations including 1x1 pointwise convolution and 3x3 depthwise convolution.

**Strengths:**

* The proposed architecture achieves better trade-off between accuracy and efficiency compared to other methods.
* When combined with other backbones, the proposed neck and head architectures achieve better performance than SSDLite.

**Weaknesses:**

* Technical contribution is limited. The basic building block of the proposed architecture is the classic inverted bottleneck structure. The paper proposed to make the first pointwise convolution optional when the input channel is the same as the expansion channel. This modification is incremental.
* Rational behind design choice is weak.
   - For the design choice that the input/output channel should not exceed 6 from the first hidden layer to the last (Ln.257), there is no theoretical proof that this leads to the minimization of inter-layer information exchange.
   - For the design choice that P4 is the core of LeYOLO's neck, there is no theoretical support but just some empirical observation.
* Presentation quality is poor. There are awkward expressions throughout the paper, e.g.,
   - Ln.318: "We improve the accuracy by using careful attention to stride details."-> by carefully designing the use of strided convolution
   - Ln.320: "in regards to computation with stridden standard convolutions" -> the use of strided convolution should be consistent.
   - Ln.431: "We have implemented a new scaling for YOLO models ...." -> a noun should follow scaling
   - Ln.436: "it seems LeYOLO is better blabla-finir".  The word "blabla-finir" is not supposed to be used here.

**Questions:**

1. Ln.224: it is not clear why C only have 4 elements given there are 5 levels and C is the output channel for each level.
2. It is not clear how the focus is put on P4 in Fig.3(c) and what is the difference between Fig.3(b) and Fig.3(c).
3. Figure 4: it is not clear what DFL represents.
4. Ln.259, it is not clear why minimizing inter-layer information exchange h_i -> h_{i+1} can be done as minimizing h1->h_n.
5. Ln.261, it is not clear why maximize I(Y;h_i) can be done by an expansion of 3 in inverted bottlenecks.

---

### Official Review · Reviewer_e9KC · 2024-10-28

**Soundness:** 2
**Presentation:** 1
**Contribution:** 2
**Rating:** 5
**Confidence:** 3

**Summary:**

This paper introduces LeYOLO, an efficient object detection model to bridges the gap between SSDLite-based object detectors and YOLO models, LeYOLO achieves a FLOP-to-accuracy ratio previously unattained, offering scalability that spans from ultra-low neural network configurations ( < 1 GFLOP) to efficient yet demanding object detection setups. Experiments on MS COCO demonstrate the efficiency and effecitvessness of LeYOLO.

**Strengths:**

- This paper proposes a new lightweight YOLO detector named LeYOLO.
- Experiments show LeYOLO is very efficient and competitive compared with other models.

**Weaknesses:**

- The presentation of this work need to be improved. The author uses too many formulas to represent the model structure, like line 215, which is not intuitive enough. It's better to use a picture to illustrate the model.
- There are many symbols not explained, making this work harder to understand. For example, $P_i$ in line 209,  $I(\cdot)$ in line 236.

minor problems:

- There are some typos in this paper, including line 188 "1x1 convolution".

**Questions:**

- It's better for the author to do more experiments to discuss the advantage of the proposed changes.
- It's better to carefully improve the written of this paper.
- The speed is often represented as FPS. It's better to replace **speed (ms)** with **latency(ms)**.

---

### Official Review · Reviewer_2RUB · 2024-11-03

**Soundness:** 1
**Presentation:** 2
**Contribution:** 1
**Rating:** 3
**Confidence:** 4

**Summary:**

This paper proposes LeYOLO, which is a CNN based architecture for object detection. Its main goal is to create a lightweight architecture in terms of FLOPs and parameter counts, yet still achieve competitive performance in terms of detection mAP. The proposed LeYOLO architecture is somewhat similar to the inverted bottleneck module proposed in MobileNet v2, with tweaks in terms of the input 1x1 convolution and striding strategy. The paper shows that LeYOLO can achieve competitive speed/accuracy tradeoff with relevant baselines such as YOLO v9, EfficientDet etc..

**Strengths:**

1. This paper is addressing an important research problem for the field of object detection, namely to identify more efficient architectures for the task. This paper has indeed proposed a novel variations of MobileNet v2 architectures which has demonstrated good accuracy/complexity (in terms of parameter count & FLOPs in particular) compared a set of recognized prior works.

2. This paper includes interesting discussions on the importance of focusing of FLOP/parameter efficiency, which is a potentially interesting read for researchers in the field. However, as noted in the weaknesses section, I do not find the discussion particularly convincing or coherent based on my understanding of the field, nor can I say that I agree with any of the conclusions.

**Weaknesses:**

This paper feels strange and confusing when I read through it. There are a few concerning issues I've seen from the paper.

## Misguided motivation
It seems to me that this paper is built on a somewhat misguided motivations. In the original word of the submissions, the research gap this paper tries to address is "there is a lack of focus on optimizing architectures based on parameter counts and FLOP in the space between state-of-the-art fast object detectors and lightweight classifiers" (Line 066-067). The conclusion, according to the submission, is that there should be a focus on "FLOP/parameter efficiency", while disregarding speed of the detectors, as discussed in detail in  "Why focus on FLOP/parameter efficiency" (Line 89 - Line 101).

I think most of the researchers in the field will find the discussions very strange, for the following reasons.

There is a rich literature of prior works on efficient object detectors. The YOLO serious of paper quoted often in the paper is one vivid example of this. All of this approach have made attempts to squeeze out more mAP with similar flops and parameter counts, making it hard to claim that there is a lack of focus on "optimizing architectures based on parameter counts and FLOP".

In the context of this paper, it seems the message the authors attempt to convey is that speed is not the only metrics used to measure how "efficient" an object detector is. I would like to acknowledge that this is thoroughly discussed in Line 076-088. This is obviously true, since a fast object detector is always a function of the underlying inference hardware. So for example, even if ShuffleNet can demonstrate better accuracy in the same FLOPs it may results in slower speed due to less efficient memory operations. What I fundamentally disagree with the authors is the conclusion from this analysis. The authors seem to suggest that, because of that, a better way to measure efficiency of an object detector architecture is through comparison with FLOPs & parameter count. But those are not very useful, precisely because they don't necessarily correlate well with one of the basic properties of a useful detector in practice - that it runs fast. Neither FLOPs nor parameter counts are useful metrics unless they happen to correlate well with real-world metrics, e.g. fast inference, low memory footprint, portable (weights are smaller hence easier to copy around). I don't think it is productive to discuss efficiency by detaching from the real-world use cases of an "efficient detector".

Whether to accept the stated motivation is particularly relevant in judging the significance of the contributions from this paper. For example, if we look closer at some empirical results. For example, in Table 11 YOLOv6n is actually a bit better than LeYOLO in QPS/mAP tradeoff, but worse in FLOPs. If we accept the claim that "FLOPs" is more relevant, then LeYOLO appears to provide a solid improvement. Otherwise, the improvement will appear much more modest, or even nonexistent.

## Incoherent empirical details
While the main contributions, as claimed by the paper, is to address the research gap that "there is a lack of focus on optimizing ... based on parameter counts and FLOP", the empirical studies are not making a strong case that LeYOLO is particularly strong in this two metrics.

For example
- Table 1 includes no comparisons in parameter counts & FLOPs, instead it is comparing a few different methods in terms of mAP and speed.
- The paper mentions in appendix (Line 1051-1054) that it was using 4GB Jetson TX2 with TensorRT to measure speed, while acknowledging that some methods in the comparison do not support TensorRT. It is unclear from the presentations which models do support TensorRT and have trustworthy results.

I would also like to mention that although mAP is used as accuracy metrics in many tables/plots in the paper, I hardly see any mention of the actual datasets used in evaluation. It is not even clear where these `mAP` numbers are coming from, e.g. it is unclear from the paper what training/validation/test sets are used to compute the mAPs.

**Questions:**

There are a lot of empirical comparisons throughout the paper, each time including a different set of methods and appear to be tested using different metrics. It will be much cleaner if a consistent set of methods are tested and if the settings (datasets, hardware/software environments when speed test is involved) are presented clearly.

---

### Meta-Review · Area_Chair_vffe · 2024-12-09

**Metareview:**

The submission presents LeYOLO, a new CNN architecture aimed at enhancing computational efficiency in object detection tasks. While LeYOLO presents an interesting approach to object detection, the current version of the paper falls short in several key areas, e.g., limited novelty, unclear motivation and presentation. The authors do not provide responses to address the above concerns, lead to a clear rejection of this submission.

**Additional Comments On Reviewer Discussion:**

The authors do not provide responses to address the above concerns.

---

### Decision · Program_Chairs · 2025-01-22

Reject